# Maximum-Likelihood Quantum State Tomography by Soft-Bayes

## Abstract

Quantum state tomography (QST), the task of estimating an unknown quantum state given measurement outcomes, is essential to building reliable quantum computing devices. Whereas computing the maximum-likelihood (ML) estimate corresponds to solving a finite-sum convex optimization problem, the objective function is not smooth nor Lipschitz, so most existing convex optimization methods lack sample complexity guarantees; moreover, both the sample size and dimension grow exponentially with the number of qubits in a QST experiment, so a desired algorithm should be highly scalable with respect to the dimension and sample size, just like stochastic gradient descent. In this paper, we propose a stochastic first-order algorithm that computes an $\varepsilon$-approximate ML estimate in $O((D \log D)/\varepsilon^2)$ iterations with $O(D^3)$ per-iteration time complexity, where $D$ denotes the dimension of the unknown quantum state and $\varepsilon$ denotes the optimization error. Our algorithm is an extension of Soft-Bayes to the quantum setup.

## 1 Introduction

Quantum state tomography (QST), the task of estimating an unknown quantum state given measurement outcomes, is essential to building reliable quantum computing devices [52]. The states of the quantum bits (qubits) prepared by an experimental apparatus are estimated, in order to check the correctness of the apparatus and, if needed, determine how to calibrate it. Moreover, quantum process tomography, the task of estimating an unknown quantum channel, can also be cast as a QST problem [7]. There are various approaches to QST, such as trace regression [49, 28, 41, 25, 63, 64], maximum-likelihood (ML) estimation [33, 34, 12], Bayesian estimation [10, 11], and recently proposed deep learning-based methods [3, 53]. Among existing approaches, the ML approach has been widely adopted for its relatively low estimation error in practice and asymptotic statistical guarantees in theory [34, 55].

Computing the ML estimator amounts to solving an optimization problem. Whereas the optimization problem is convex, standard convex optimization methods are not directly applicable. It is easily checked that the negative log-likelihood function in ML QST is neither Lipschitz nor smooth, violating standard assumptions in optimization literature [39, 24]. Hence, for example, even whether vanilla gradient descent converges for QST is unclear. This is perhaps why $R\rho R$, a heuristic algorithm known to be empirically fast, was developed via an expectation maximization, instead of convex optimization, argument [43, 44]. Unfortunately, $R\rho R$ does not always converge [60]. The negative log-likelihood function is indeed self-concordant, so Newton's method is readily applicable [45]. Nevertheless, the dimension of a quantum state grows exponentially with the number of qubits; the Hessian computations in Newton's method are computationally too expensive when the dimension is high. There are a few first-order (i.e., gradient-based) convex optimization algorithms

Submitted to 36th Conference on Neural Information Processing Systems (NeurIPS 2022). Do not distribute.

provably converging for ML QST, such as diluted $R\rho R$ [60, 27], SCOPT [57][1], NoLips [9], the Frank-Wolfe method [48, 24, 65, 14], and entropic mirror descent with line search [39]. These are all batch methods. As they require computing the full gradient in every iteration, their per-iteration time complexities are at least linear in the sample size. To estimate a quantum state, it has been proved that the sample size must be exponential in the number of qubits [47, 30, 17].

Regarding the high dimension and sample size issues in ML QST, it is desirable to, like how we handle the same issues in modern machine learning applications, develop a stochastic first-order optimization method for ML QST. A stochastic first-order optimization method takes one or a few, instead of all, samples in each iteration and avoids computationally expensive Hessian computations. The stochastic quasi-Newton method for self-concordant minimization of Zhou et al. [66] seems to apply. Nevertheless, its step size selection rule involves Hessian computations; moreover, its analysis assumes a bounded Hessian, which does not hold in ML QST. The stochastic mirror-prox and stochastic primal-dual hybrid gradient methods were considered for problems very similar to ML QST [4, 16, 32]. However, their analyses assume either a bounded dual domain or Lipschitzness; both are violated in ML QST.

In this paper, we propose a stochastic first-order algorithm for ML QST. We design the algorithm by an online learning argument. Consider an online convex optimization problem, where the loss function in each round corresponds to the negative log-likelihood function corresponding to one data point in ML QST. Interestingly, this online convex optimization problem is exactly the quantum analogue of online portfolio selection, a celebrated online learning problem [19, 20]. Since the ML approach aims to minimize the empirical average of the negative log-likelihood, once we "quantumize" any existing first-order online portfolio selection algorithm that is no-regret and apply an online-to-batch conversion [15, 22], the resulting algorithm becomes a stochastic first-order algorithm for ML QST. We refer the reader to Section 2 for an introduction of relevant concepts.

The algorithm we choose to "quantumize" is Soft-Bayes [51]. There are two reasons. First, the per-round time complexity of Soft-Bayes is linear in the ambient dimension, arguably the lowest one can expect; second, Soft-Bayes has a curious connection with expectation maximization [43, 44] (see Section 5). We call the resulting algorithm Stochastic Q-Soft-Bayes. Stochastic Q-Soft-Bayes processes one randomly chosen data point in each iteration. Suppose the quantum state to be estimated is represented by a $D$-by-$D$ density matrix. The per-iteration time complexity of Stochastic Q-Soft-Bayes is $O(D^3)$, independent of the sample size. The expected optimization error of Stochastic Q-Soft-Bayes converges to zero at a $O(\sqrt{(1/T)D\log D})$ rate, where $T$ denotes the number of iterations.

The main technical difficulty lies in figuring out an appropriate quantum extension of Soft-Bayes that coincides with Soft-Bayes when all matrices involved share the same eigenspace and allows for a regret analysis. This is challenging because for any given "non-quantum" expression, one can immediately find many candidates for its quantum extension, but only a few or one of them inherit the desired theoretical properties of their "non-quantum" counterpart; see, e.g., the discussion in [62, Chapter 11] for extending information theoretic quantities to the quantum case. Similar to the quantum extension of exponentiated gradient update by Tsuda et al. [58], the quantum extension we find reveals the complicated mathematical structure of Soft-Bayes hidden in the "non-quantum" setup.

Instead of empirically beating state of the arts, our aim is to give the first provably fast stochastic first-order algorithm for ML QST. Section 3.3 shows that Stochastic Q-Soft-Bayes is competitive in time complexity in comparison to existing batch algorithms. Section 4 shows that Stochastic-Soft-Bayes is empirically even faster than $R\rho R$ in terms of the number of epochs. Unfortunately, Section A shows that in terms of the elapsed time, Q-Soft-Bayes may not be satisfactory to practitioners. We discuss the possibility of developing faster stochastic first-order methods in Section 5.

## 1.1 Related work

A textbook approach to quantum state tomography is to approximate the problem as a trace regression problem [46] and compute the corresponding least-squares estimate or directly minimize the expected square loss, sometimes with regularization [49, 28, 25, 64, 63]. Since minimizing the square loss is arguably the most standard problem in optimization and machine learning, many existing algorithms apply. Youssry et al. [64] proved the convergence of stochastic entropic mirror descent. Yang et al.

---

[1]A similar algorithm is proposed and studied in [26], but the bounded Hessian assumption therein renders the algorithm inapplicable to ML QST.

[63] showed that several standard online learning algorithms are no-regret for the corresponding online trace regression problem. Notice that both papers do not consider the ML formulation.

Quantum state tomography schemes optimal or nearly optimal in sample complexity are known [47, 30, 38, 29]. The optimal schemes require entangled measurements, challenging to implement [47, 30]. If only incoherent measurements (as in the ML QST scheme considered in this paper) are allowed, the scheme by Kueng et al. [38] is optimal [17]; nevertheless, the scheme is still challenging to implement [38, p. 97]. The scheme proposed by Guţă et al. [29] is nearly optimal, but the numerical result in [29, Figure 1] shows that the ML approach achieves a smaller estimation error empirically.

A problem closely related to quantum state tomography is shadow tomography, in which one is not interested in recovering the quantum state but estimating the probability distributions of its measurement outcomes [1]. Aaronson et al. showed that shadow tomography can be done in an online fashion, via follow the regularized leader with the von Neumann entropy [2]. We emphasize that shadow tomography is fundamentally different from quantum state tomography. Indeed, Aaronson showed shadow tomography is strictly easier than state tomography, in the sense that the former requires much less samples than the latter [1]. Another closely related problem is the quantum version of individual sequence prediction considered by Koolen et al. [37]. The loss function studied in [37] is the trace-log loss, instead of the log-trace loss we consider, as discussed in Section 4 of their paper.

Our algorithm is developed via "quantumizing" an online portfolio selection algorithm. Online portfolio selection is a classic online learning problem. It is known that the optimal regret of online portfolio selection is $O(D \log T)$, where $D$ denotes the ambient dimension and $T$ denotes the number of rounds, and is achieved by Universal Portfolio Selection (UPS) [19, 20]. However, UPS is computationally too expensive to be practical [35]. There are several algorithms that try to balance between the regret and computational complexity, but none of them is optimal in both aspects [42, 59]. Soft-Bayes strikes a balance with a $O(D)$ per-round time complexity and $O(\sqrt{TD \log D})$ regret.

Recently, Zimmert et al. [67] "quantumized" another online portfolio selection algorithm, called BISONS, to solve the game of online quantum state tomography described in Section 3.1[2]. By an online-to-batch conversion, their algorithm yields a stochastic algorithm for ML QST. The resulting algorithm achieves a better iteration complexity than Stochastic Q-Soft-Bayes; nevertheless, each iteration of it requires solving a self-concordant convex program by, e.g., Newton's method, resulting in a high time complexity incomparable to that of Stochastic Q-Soft-Bayes. In the words of Zimmert et al. [67], both their and our algorithms are on the state-of-the-art efficiency-regret frontier.

## 1.2 Notations

We write $\mathbb{R}_+$ for the set of non-negative real numbers and $\mathbb{R}_{++}$ the set of strictly positive real numbers. Let $J \in \mathbb{N}$. We write $[J]$ for the set $\{1, \ldots, J\}$. Let $M$ be a matrix. We write $M^H$ for its Hermitian (conjugate transpose) and $\operatorname{tr}(M)$ for its trace. Let $H$ be a Hermitian matrx; we write its spectral decomposition as $H = \sum_d \lambda_d P_d$, where $\lambda_d$ are the eigenvalues and $P_d$ are projections onto the associated eigenspaces. Let $f$ be a real-valued function whose domain contains $\{\lambda_d\}$. Then, $f(H)$ is defined as the matrix $\sum_d f(\lambda_d) P_d$. Let $A$ and $B$ be two matrices. We write $A \geq B$ if and only if $A - B$ is positive semi-definite. Let $\mathcal{E}$ be an event and $\xi$ be a random variable following a probability distribution $P$. We write $\mathsf{P}(\mathcal{E})$ for the probability of the event and $\mathsf{E}_P[\xi]$ for the expectation of $\xi$. We sometimes omit the subscript $P$ and write $\mathsf{E}[\xi]$ when there is no ambiguity.

## 2 Preliminaries

### 2.1 Maximum-Likelihood Quantum State Tomography

In the mathematical formulation of quantum mechanics, a quantum state corresponds to a *density matrix*, a Hermitian positive semi-definite complex matrix of unit trace. Let the dimension of the density matrix be $D \in \mathbb{N}$. If there are $q$ qubits, then $D = 2^q$. We denote by $\mathcal{D}$ the set of density matrices in $\mathbb{C}^{D \times D}$, i.e.,

$$\mathcal{D} := \left\{ \rho \mid \rho \in \mathbb{C}^{D \times D}, \rho = \rho^H, \rho \geq 0, \operatorname{tr} \rho = 1 \right\}.$$

---

[2]We note that the work of Zimmert et al. [67] appears much later than the arXiv version of our work. This footnote is simply to address potential confusions of reviewers and may be removed in the camera-ready version. **We do not encourage the reviewers to check the arXiv version as that violates the double-blind policy.**

A measurement setup corresponds to a *positive operator-valued measure (POVM)*, a set of Hermitian positive semi-definite complex matrices summing to the identity. Let $\rho \in \mathcal{D}$ and $\{ M_1, \ldots, M_J \} \subset \mathbb{C}^{D \times D}$ be a POVM. The measurement outcome is a random variable $\eta$ taking values in $[J]$ such that

$$\mathsf{P}\,(\eta = j) = \mathrm{tr}(M_j \rho), \quad \forall j \in [J].$$

The ML estimation approach seeks the quantum state that maximizes the probability of observing the measured data. Let $\rho^\natural \in \mathcal{D}$ be the density matrix to be estimated. In a standard QST experiment, we construct $N$ independent copies of $\rho^\natural$ and measure the copies independently with possibly different POVMs. It is easily checked that the ML estimator is of the form

$$\hat{\rho} \in \operatorname*{argmax}_{\rho \in \mathcal{D}} \prod_{n=1}^{N} \mathrm{tr}(A_n \rho),$$

where each $A_n$ is an element of the POVM for the $n$-th measurement. We call $\{ A_1, \ldots, A_N \}$ the *data-set*. We equivalently write the ML estimator as

$$\hat{\rho} \in \operatorname*{argmin}_{\rho \in \mathcal{D}} f(\rho), \tag{1}$$

$$f(\rho) := \frac{1}{N} \sum_{n=1}^{N} \left( -\log \mathrm{tr}(A_n \rho) \right). \tag{2}$$

Obviously, (1) is a convex optimization problem. If the matrix $A_{n'}$ is not full-rank for some $n' \in [N]$ (as in the cases with the Pauli measurement [40] and Pauli basis measurement [54, 56]), then $\mathrm{tr}(A_{n'}\rho)$ can be arbitrarily close to zero on $\mathcal{D}$ and hence the $k$-th-order derivative of the objective function $f$ is unbounded for all $k \in \mathbb{N}$.

Let $A$ be a random matrix following the empirical distribution $\hat{P}_N$ on the data-set $\{ A_1, \ldots, A_N \}$. If the matrices $A_n$ are all different, then $\hat{P}_N$ is simply the uniform distribution on the data-set $\{ A_1, \ldots, A_N \}$. Then, we can write the objective function in (1) as an expectation

$$f(\rho) = \mathsf{E}_{\hat{P}_N} \left[ -\log \mathrm{tr}(A\rho) \right]. \tag{3}$$

This observation connects ML QST with the problem of computing the log-optimal portfolio.

## 2.2 Log-optimal Portfolio

Interestingly, the optimization problem (1) is exactly a quantum extension of the problem of computing the log-optimal portfolio (aka the Kelly criterion), an asymptotically optimal strategy for long-term investment [5, 13, 36]. Consider multi-round investment in a market. Suppose there are $D$ investment alternatives. For the $t$-th round, we list the return rates of the investment alternatives in that round as a random vector $a_t \in \mathbb{R}_+^D$. Before each round starts, an investor needs to determine the portfolio for the round given the past return rates. Denote by $P_{t+1}$ the probability distribution of $a_{t+1}$ conditional on the history $(a_1, \ldots, a_t)$. The log-optimal portfolio $w_{t+1}^\star$ for the $(t+1)$-th round is given by the stochastic optimization problem:

$$w_{t+1}^\star \in \operatorname*{argmin}_{w \in \Delta} \varphi(w), \tag{4}$$

$$\varphi(w) := \mathsf{E}_{P_{t+1}} \left[ -\log \langle a_{t+1}, w \rangle \right], \tag{5}$$

where $\Delta$ denotes the probability simplex in $\mathbb{R}^D$, the set of entry-wise non-negative vectors whose entries sum to one. Then, the investor distributes the wealth to the investment alternatives following the ratios specified by $w_{t+1}^\star$.

We now discuss the correspondence between ML QST and log-optimal portfolio. The set $\mathcal{D}$ is indeed a quantum extension of the probability simplex $\Delta$, in the sense that a Hertimian matrix is a density matrix if and only if its vector of eigenvalues lies in the probability simplex. The objective functions in (1) and (4) are both expectations of the logarithm of linear functions. Indeed, it is easily checked that if the matrices involved in (1) share the same eigenbasis, then the non-commutativity issue in the quantum setup vanishes and (1) coincides with (4). Though the correspondence is obvious given the two problem formulations, it seems that this correspondence has not been discussed in the literature.

## 2.3 Online Portfolio Selection

Online portfolio selection may be viewed as a probability-free version of log-optimal portfolio [19]. Online portfolio selection is a multi-round game between two players, say INVESTOR and MARKET. Suppose the game consists of $T$ rounds. In the $t$-th round of the game, first, INVESTOR announces a portfolio $w_t \in \Delta$; then, MARKET announces the return rates of all investment alternatives for the $t$-th round in a vector $a_t \in \mathbb{R}_+^D$; finally, INVESTOR suffers a loss of value $-\log \langle a_t, w_t \rangle$. The goal of IN-VESTOR is to achieve a low *regret* against all possible strategies of MARKET. The regret is defined as

$$R_T := \sup \sum_{t=1}^{T} \left( -\log \langle a_t, w_t \rangle \right) - \min_{w \in \Delta} \sum_{t=1}^{T} \left( -\log \langle a_t, w \rangle \right),$$

where the supremum is over all possible strategies of MARKET to determine $(a_t)_{1 \leq t \leq T}$. We say an algorithm for INVESTOR to determine the portfolios is *no-regret* if it achieves $R_T = o(T)$.

If we can sample from the conditional probability distribution $P_{t+1}$ specified in the previous sub-section, then we can transform a no-regret online portfolio selection algorithm to an algorithm that approximately computes the log-optimal portfolio. The following is an immediate consequence of the online-to-batch conversion [15, 50].

**Proposition 2.1.** *Suppose in the online portfolio selection game, the vectors $a_t$ are all independent and identically distributed (i.i.d.) random vectors following the probability distribution $P_{t+1}$ in the previoius sub-section. Let $(w_t)_{t \in \mathbb{N}}$ be the sequence of iterates generated by an algorithm for INVESTOR of regret $R_T$. Then, for any $T \in \mathbb{N}$,*

$$\mathsf{E}\left[ \varphi(\overline{w}_T) - \min_{w \in \Delta} \varphi(w) \right] \leq \frac{R_T}{T}, \quad \overline{w}_T := \frac{w_1 + \cdots + w_T}{T}.$$

*Recall that $\varphi$ is the conditional expectation of the log-linear loss in* (5).

If INVESTOR adopts a no-regret algorithm, then the expected optimization error vanishes as $T \to \infty$.

## 2.4 Soft-Bayes

There are various existing algorithms for online portfolio selection. Among these algorithms, we are particularly interested in Soft-Bayes [51]. The per-iteration time complexity of Soft-Bayes is linear in $D$, arguably the lowest one can expect. This is a desirable feature for ML QST, as the dimension of the density matrix grows exponentially with the number of qubits.

The iteration rule of Soft-Bayes is as follows.

- Initialize at $w_1 = (1/D, \ldots, 1/D) \in \Delta$ (the uniform distribution).
- For each $t \in \mathbb{N}$, compute

$$w_{t+1} = (1 - \eta)w_t + \eta \frac{a_t \circ w_t}{\langle a_t, w_t \rangle}, \quad \forall t \in \mathbb{N}, \tag{6}$$

for some properly chosen *learning rate* $\eta \in [0, 1]$, where $\circ$ denotes the entry-wise product.

Soft-Bayes has the following regret guarantee.

**Theorem 2.2** ([51]). *After $T$ rounds in online portfolio selection, the regret of Soft-Bayes with*

$$\eta = \frac{\sqrt{DT}}{\sqrt{DT} + \sqrt{\log D}} \tag{7}$$

*is at most $2\sqrt{TD \log D} + \log D$.*

## 3 Online Maximum-Likelihood Quantum State Tomography by Q-Soft-Bayes

Following the discussion in Section 1, we first propose a *game of online quantum state tomography* as a quantum extension of online portfolio selection. Then, we "quantumize" Soft-Bayes to derive a no-regret algorithm for the game and analyse its regret. Finally, we adopt the online-to-batch conversion and bound the expected optimization error of the resulting algorithm.

## 3.1  Game of Online Quantum State Tomography

We propose the following game of online quantum state tomography as a quantum extension of online portfolio selection. Online quantum state tomography is a multi-round game between two players, say PHYSICIST and ENVIRONMENT. Suppose there are in total $T$ rounds. In the $t$-th round, first, PHYSICIST announces a density matrix $\rho_t \in \mathcal{D}$; then, ENVIRONMENT announces a Hermitian positive semi-definite matrix $A_t \geq 0$; finally, PHYSICIST suffers for a loss of value $-\log \operatorname{tr}(A_t \rho_t)$. The regret in this game is given by

$$R_T := \sup \sum_{t=1}^{T} \left( -\log \operatorname{tr}(A_t \rho_t) \right) - \min_{\gamma \in \mathcal{D}} \sum_{t=1}^{T} \left( -\log \operatorname{tr}(A_t \rho) \right),$$

where the supremum is over all possible strategies of PHYSICIST to generate the sequence $(A_t)_{1 \leq t \leq T}$.

The connection with online portfolio selection is obvious and similar to that between ML QST and the log-optimal portfolio: The vector of eigenvalues of a density matrix lies in the probability simplex $\Delta$; the Hermitian matrices $A_t$ and the positive semi-definiteness condition correspond to the vectors $a_t$ in online portfolio selection and their non-negativity condition, respectively; the losses in the two games are both logarithms of linear functions; the regrets in the two games are defined exactly in the same manner. When all the matrices involved in the game of online quantum state tomography share the same eigenbasis, we recover the game of online portfolio selection.

## 3.2  Q-Soft-Bayes and Stochastic Q-Soft-Bayes

We propose the following Q-Soft-Bayes algorithm as a quantum extension of Soft-Bayes.

- Initialize at $\rho_1 = W_1 = I/D$.

- For each $t \in \mathbb{N}$, compute

$$\begin{aligned}
G_t &= (1 - \eta)I + \eta \frac{A_t}{\operatorname{tr}(A_t \rho_t)}, \\
W_{t+1} &= \exp\left( \log\left(W_t\right) + \log\left(G_T\right) \right), \\
\rho_{t+1} &= \frac{W_{t+1}}{\operatorname{tr}(W_{t+1})},
\end{aligned} \tag{8}$$

for some properly chosen learning rate $\eta \in [0, 1]$.

*Remark* 3.1. Recently, we learned that Q-Soft-Bayes may be interpreted using the *commutative matrix product* by Warmuth and Kuzmin [61]. It is currently unclear to us whether this interpretation provides any insight.

If we were able to cancel the exponential and logarithms in Q-Soft-Bayes, then we recover Soft-Bayes; however, due to the non-commutativity issue, such cancellation is illegal in general. In comparison to Soft-Bayes, Q-Soft-Bayes has an additional normalization step to ensure its outputs are of unit trace. We prove the following in Section B.

**Proposition 3.2.** *It holds that* $\operatorname{tr}(W_t) \leq 1$ *for all* $t$.

Numerical experiments show that the equality does not always hold, so the normalization step is necessary. Recall that Soft-Bayes does not need the normalization step (see Section 2.4).

In Appendix C, we prove the following regret bound for Q-Soft-Bayes, showing that it inherits the regret bound of Soft-Bayes.

**Theorem 3.3.** *The regret of Q-Soft-Bayes with the learning rate* $\eta$ *given in* (7) *is at most* $2\sqrt{TD \log D} + \log D$.

*Remark* 3.4. One might wonder why $D$ is not replaced by $D^2$ in the quantum case. This is because in our extension, the analogue of a $D$-dimensional vector in the quantum case is the $D$-dimensional vector of eigenvalues of a $D$-by-$D$ Hermitian matrix, instead of the $D$-dimensional vector obtained by vectorizing a $\sqrt{D}$-by-$\sqrt{D}$ matrix. Similar coincidence of regret bounds can be observed in, for example, the matrix version of exponentiated gradient update [58, 8] and the quantum individual sequence prediction algorithms of Koolen et al. [37].

The standard online-to-batch conversion argument can also be applied to solving ML QST by Q-Soft-Bayes. Recall for ML QST, our aim is to solve the stochastic optimization problem

$$\hat{\rho} \in \underset{\rho \in \mathcal{D}}{\operatorname{argmin}} \, \mathsf{E}_{\hat{P}_N} \left[ -\log \operatorname{tr}(A, \rho) \right],$$

where $A$ is a random matrix following the empirical probability distribution $\hat{P}_N$ on the data-set $\{ A_1, \ldots, A_N \}$ (see Section 2.1). We propose the following stochastic optimization algorithm, which we call Stochastic Q-Soft-Bayes, to solve ML QST.

- Initialize Q-Soft-Bayes with $\rho_1 = W_1 = I/D$.

- In the $t$-th iteration of Stochastic Q-Soft-Bayes, do the following.

    1. Output the $t$-th output $\rho_t$ of Q-Soft-Bayes.
    2. Sample a random matrix $B_t \in \{ A_1, \ldots, A_N \}$ following the empirical probability distribution $\hat{P}_N$ on the data-set, independent of the past.
    3. Let ENVIRONMENT in the online QST game announce the matrix $B_t$.

Similarly as for Proposition 2.1, the standard online-to-batch conversion argument provides the following convergence guarantee of Stochastic-Q-Soft-Bayes.

**Proposition 3.5.** *Let $(\rho_t)_{t \in \mathbb{N}}$ be the sequence of iterates generated by Stochastic Soft-Bayes. Then, for any $T \in \mathbb{N}$, it holds that*

$$\mathsf{E} \left[ f(\overline{\rho}_T) - \min_{\rho \in \mathcal{D}} f(\rho) \right] \leq 2\sqrt{\frac{D \log D}{T}} + \frac{\log D}{T},$$

*where $\overline{\rho}_T \coloneqq (\rho_1 + \cdots + \rho_T)/T$ and the expectation is with respect to the randomness in $B_t$ of Stochastic Soft-Bayes. Recall $f$ is the objective function in ML QST as defined in (2) or (3) (the two definitions are equivalent).*

Therefore, Stochastic-Q-Soft-Bayes outputs an approximate ML estimator of expected optimization error smaller than $\varepsilon$ in $O((D \log D)/\varepsilon^2)$ iterations. Each iteration of Stochastic-Q-Soft-Bayes requires computing a matrix exponential and two matrix logarithms. The overall time complexity is hence $O((D^4 \log D)/\varepsilon^2)$. One may adopt anytime online-to-batch [22], which seems to empirically yield faster convergence. According to [22], the optimization error guarantee remains the same; the only difference is that $\nabla f$ are evaluated at $\overline{\rho}_t$ instead of $\rho_t$ when implementing Soft-Bayes, so the overall time complexity also remains the same.

One may be interested in the distance to the minimizer. It is easily checked that the function $f$ is self-concordant. If $\nabla^2 f$ is positive definite at the minimizer, a standard condition for well-posed estimators, then the function $f$ is locally strongly convex around the minimizer [45, Theorem 4.1.6]. Therefore, the distance to the minimizer, measured in terms of the Frobenius norm, is asymptotically of the order of the square root of the optimization error.

### 3.3 Theoretical Comparison with Existing Batch Algorithms

Let us compare the time complexities of Stochastic Q-Soft-Bayes and existing algorithms discussed in Section 1. The iteration complexities of existing algorithms are mostly unknown or vague in their dependence on the problem parameters. Diluted $R\rho R$ and entropic mirror descent with line search do not have non-asymptotic analysis results [60, 27, 39]; SCOPT only has a local linear rate guarantee [57]; Adaptive Frank-Wolfe and Monotonous Frank-Wolfe have $O(\varepsilon^{-1})$ iteration complexities with unclear dependence on the dimension and sample size, as their error bounds involve local smoothness parameters that are hard to evaluate [14, 24]. A finer analysis of Adaptive Frank-Wolfe by Zhao and Freund [65] shows that its iteration complexity is $O(\varepsilon^{-1} N)$ and hence its time complexity is $O\left(\varepsilon^{-1}(N^2 D^2 + N\tau)\right)$, where the symbol $\tau$ denotes the time of computing the local norm defined by the Hessian, for which we do not know an efficient implementation. In comparison, the complexities of Stochastic Q-Soft-Bayes is very clear: $O(\varepsilon^{-2} D \log D)$ iteration complexity and hence $O(\varepsilon^{-2} D^4 \log D)$ time complexity. The time complexity of Stochastic Q-Soft-Bayes becomes competitive with Adaptive Frank-Wolfe if $N \gg D\sqrt{(1/\varepsilon) \log D}$, *ignoring the time of computing the local norms.* Recently, it is proved that any QST scheme with non-coherent measurement, e.g., ML QST we consider in this paper, requires $N = \Omega(D^3/\delta^2)$ to achieve an estimation error smaller than $\delta$

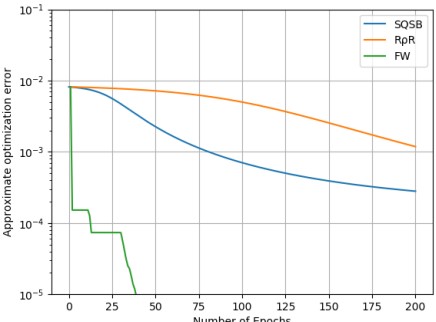
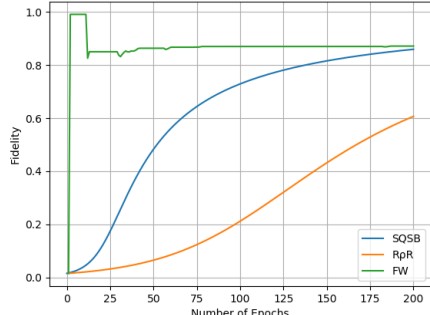

Figure 1: Approximate optimization errors in function value of Stochastic Q-Soft-Bayes (SQSB), $R\rho R$ ~~(RrhoR)~~, and Monotonous Frank-Wolfe (FW).

Figure 2: Fidelity values of the iterates and the W state achieved by Stochastic Q-Soft-Bayes (SQSB), $R\rho R$ ~~(RrhoR)~~, and Monotonous Frank-Wolfe (FW).

in the trace distance [17]. The algorithm by Zimmert et al. [67] has a $\tilde{O}(D^3/\varepsilon)$ iteration complexity and $O(D^6)$ per-iteration time complexity ignoring the dependence on other parameters, due to the use of Newton's method to compute the iterates; the overall time complexity has a much higher dependence on the dimension than Adaptive Frank-Wolfe and Stochastic Q-Soft-Bayes. We conclude that the time complexity of Stochastic Q-Soft-Bayes is competitive compared to existing algorithms.

## 4 Numerical Results

As discussed above, Stochastic Q-Soft-Bayes is competitive in theory. We now examine its empirical performance with anytime online-to-batch. We compare its empirical speed with two batch methods, the $R\rho R$ method [43, 44] and Monotonous Frank-Wolfe [14], on a synthetic data-set in Figure 1 and Figure 2. We have mentioned several batch methods applicable for ML QST in Section 1. Among them, we choose $R\rho R$ for comparison as it is representative in physics literature and empirically fast, though *it does not always converge*. We choose monotonous Frank-Wolfe for comparison as it avoids computationally expensive Hessian computations in step size selection. Recall that Monotonous Frank-Wolfe converges at a $O(1/t)$ rate as other Frank-Wolfe methods for self-concordant minimization do [65, 24, 48], but its complexity guarantee lacks a clear characterization of the dependence on the dimension and sample size.

The synthetic data-set is generated basically following the set-up in [31]. The number of qubits $q$ equals 6. The dimension $D$ then equals $2^q = 64$. The unknown quantum state to be measured is the W state. We randomly generate $N = 4^q \times 100 = 409600$ Pauli observables as in, e.g., [25, 28, 40], each of which corresponds to a POVM of two rank-$(D/2)$ elements. As there are in total $4^q$ different Pauli observables (and hence POVMs), each observable is used about 100 times. Then, we sample the $N$ measurement outcomes and formulate the ML estimator following Section 2.1.

The performance measures we consider are optimization errors (in objective function) and fidelity values. To estimate the optimization error, we run each algorithm for 200 epochs and use the smallest function value found by the algorithms as an approximate optimal value. The approximate optimization error of an iterate is defined as the difference between the objective function value at the iterate and the approximate optimal value. Fidelity is a notion commonly used by physicists to measure how close two quantum states are to each other. For any two density matrices $\rho$ and $\sigma$, the fidelity is given by $F(\rho, \sigma) := \left(\text{tr}\sqrt{\sqrt{\rho}\sigma\sqrt{\rho}}\right)^2$, which takes values in $[0, 1]$. The fidelity of two quantum states equals 1, if the two states are exactly the same. We plot the optimization errors and fidelity values versus the number of epochs. An epoch corresponds to one pass of the whole data-set. One iteration of Stochastic Q-Soft-Bayes corresponds to $1/N$ epoch. One iteration of $R\rho R$ and Monotonous Frank-Wolfe corresponds to 1 epoch as both algorithms are batch.

Obviously, Stochastic Q-Soft-Bayes converges faster than $R\rho R$ in both optimization error and fidelity. Where as Monotonous Frank-Wolfe is the fastest in both figures, this can be explained by the fact that

Frank-Wolfe tends to generate approximately low-rank iterates. The W state corresponds to a rank-1 density matrix, so the ML estimate should be approximately low-rank, matching the structure favoured by Frank-Wolfe. We conclude that the convergence speed of Stochastic Q-Soft-Bayes is competitive in theory (Section 3.3) and comparable to fast yet theoretically non-rigorous algorithms in practice.

A comparison in terms of the elapsed time is provided in Appendix A. The results show there is a large room for improvement to compete with $R\rho R$ and Monotonous Frank-Wolfe in the elapsed time. The source codes are provided in the supplementary material.

## 5 Discussions

### 5.1 Can We Find a Faster Stochastic First-Order Algorithms for ML QST?

Our approach to constructing a stochastic first-order algorithm for ML QST conceptually applies to any no-regret online portfolio selection algorithm. In this paper, we focus on Soft-Bayes. Other existing online portfolio selection algorithms have much higher per-iteration time complexities, in terms of the dependence on the ambient dimension and sample size. If we adopt any other existing online portfolio selection algorithm and "quantumize" it to obtain a stochastic algorithm for ML QST, then the resulting algorithm will scale poorly with the number of qubits. Developing an online portfolio selection algorithm that enjoys both a low regret and low time complexity is still open [59, 67].

It is still possible to develop another quantum extension of Soft-Bayes that enjoys a lower per-iteration time complexity. The per-iteration time complexity issue may be mitigated if we consider other quantum extensions of Soft-Bayes. For example, if we naïvely replace (8) by $W_{t+1} = (G_t W_t + W_t G_t)/2$, the resulting algorithm still coincides with Soft-Bayes when all matrices share the same eigenbasis, whereas the per-iteration time complexity is reduced to $O(D^\omega)$ for some $\omega < 2.373$ [6]. Unfortunately, we cannot work out a non-asymptotic analysis for any other possible quantum extension of Soft-Bayes we can think of.

The discussion above assumes that we adopt the online-to-batch argument as in this paper. Another way, which we think perhaps more plausible, is to directly consider the stochastic optimization formulation and develop a stochastic optimization algorithm for ML QST.

### 5.2 Connection with Expectation Maximization

Finally, let us discuss an interesting connection between Q-Soft-Bayes and expectation maximization (EM). The $R\rho R$ algorithm, according to [43, 44], was inspired by the expectation maximization (EM) method for solving optimization problems of the form (4). Given a full-rank initial iterate $\rho_1 \in \mathcal{D}$, $R\rho R$ iterates as

$$\rho_{t+1} = \frac{R_t \rho_t R_t}{\mathrm{tr}(R_t \rho_t R_t)}, \quad R_t := -\nabla f(\rho_t), \quad \forall t \in \mathbb{N},$$

where $f$ is defined in (2). In comparison, given an entry-wise positive vector $w_1 \in \Delta$, EM for (4) iterates as

$$w_{t+1} = w_t \circ (-\nabla\varphi(w_t)), \quad \forall t \in \mathbb{N}.$$

It is interesting to notice that even when all matrices involved share the same eigenbasis, $R\rho R$ is not equivalent to EM. Indeed, EM is proved to asymptotically converge to the optimum [18, 21], whereas $R\rho R$ oscillates on a carefully designed data-set [60]. This suggests that $R\rho R$ is perhaps not a "natural" quantum extension of EM. Later, there were variations of $R\rho R$ that solve the convergence issue by line search [60, 27], but these variations still do not recover EM.

Notice that the formulation of Soft-Bayes (6) is the convex combination of the previous iterate and *the output of EM*. Therefore, Soft-Bayes, after the online-to-batch conversion, can be interpreted as a relaxed stochastic EM method for computing the log-optimal portfolio. As Q-Soft-Bayes becomes Soft-Bayes when all matrices involved share the same eigenbasis, we may claim that Stochastic Q-Soft-Bayes is also a relaxed stochastic EM method, though its derivation does not have any obvious relation with the standard derivation of EM [23].

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
