# OpenReview forum: "Maximum-Likelihood Quantum State Tomography by Soft-Bayes"
_NeurIPS.cc/2022/Conference — NeurIPS 2022 Submitted_

### Official Review · Reviewer_oBoV · 2022-07-08

**Rating:** 3
**Confidence:** 4
**Soundness:** 2 fair
**Presentation:** 2 fair
**Contribution:** 2 fair

**Summary:**

In this paper the authors consider the problem of quantum state tomography (QST). Here there is an unknown n qubit quantum state rho (i.e., its a complex matrix of size 2^n x 2^n) and the goal is to learn rho given copies to the state. The sample complexity of QST has completely been pinned down to Theta(d^2) where d=2^n. Apart from the theoretical guarantees there are also some heuristic works that have looked at QST and one such method which is relevant to this work is that of maximum-likelyhood estimation. So computing the ML estimator  amounts to solving a convex optimization problem, however the main issue for using maximum likelehood estimator for QST is  negative log-likelihood function in ML QST is not Lipschitz and not smooth as well.  In contrast in this paper, the authors develop a stochastic first-order optimization technique for ML QST inspired by techniques from online learning. They moreover choose to quantumize  the soft-Bayes algorithm which is an iterative algorithm  whose round by round complexity is linear in the dimension of the state, i.e., d  and the rate of convergence of the procedures presented in this paper are similar to ones known from literature. Finally the authors provide some empirical plots implementing their learning procedures.

I think the study of QST using MLE is nice, but i dont think this paper is fit for NeurIPS for a few reasons
a) The main contributions of this paper are in a very narrow subject and i dont think this narrow subject  might be of interest to classical ML (which is the primary audience of NeurIPS) and quantum computing (since the novelty of this paper is limited to improving certain heuristic-based approach for QST).
b) In my opinion, the main contributions of this paper needs more motivation. For example, QST is a completely solved subject, in the sense the sample complexity has been pinned down. Then why should one look at solving QST using regression, MLE etc? Is there any solid motivation for considering these approaches?

Given the niche contributions which are limited in applicability, i think this paper isnt strong enough to be accepted for NeurIPS.

**Questions:**

Mentioned earlier.

**Ethics Review Area:**

["Responsible Research Practice (e.g., IRB, documentation, research ethics)"]

**Limitations:**

Mentioned earlier.

**Strengths And Weaknesses:**

Mentioned earlier.

---

> ### Author Response · Authors · 2022-07-29
> **On the relevance**
>
> Dear Reviewer oBoV,
>
> Thanks for your comments. If we understand correctly, the main concern is the relevance of our result. We address below the reasons behind the concerns.
>
> - *``The main contributions of this paper are in a very narrow subject and i dont think this narrow subject might be of interest to classical ML  (which is the primary audience of NeurIPS).''*
>
>     **We do not agree.** In online learning, for example, Warmuth and his collaborators have contributed several papers on quantum extensions of classical results, such as [55], [58], [35] cited in our paper. Two papers on online learning quantum states with Lipschitz losses, cited as [2] and [60] in our paper, were published in NeurIPS 2018 and AAAI 2020, respectively; an adaptive extension ("Adaptive online learning of quantum states" by Chen et al.) appeared after the submission deadline this year. **Most notably, Zimmert et al. (cited as [64] in our paper) considered a quantum extension of their online portfolio selection algorithm to study exactly the same online learning problem as in our paper; the paper was just published in this year's COLT.**
>
> - *``the novelty of this paper is limited to improving certain heuristic-based approach for QST''*
>
>     Our aim is not in improving the heuristic $R \rho R$ but proposing a (relatively) computationally efficient algorithm for ML QST. In Section 3.3, **we have provided an exhaustive theoretical comparison with all algorithms we know that have rigorous complexity guarantees** and conclude the competitiveness of our algorithm in theory. In Section 4, we compare our algorithm with $R \rho R$ and Frank-Wolfe, simply because other algorithms are computationally too expensive to implement.
>
> -  *``For example, QST is a completely solved subject, in the sense the sample complexity has been pinned down. Then why should one look at solving QST using regression, MLE etc? Is there any solid motivation for considering these approaches? ''*
>
>     We would like to emphasize that **our aim is to present an efficient algorithm to compute the ML estimate, not to improve the sample complexity.** Some notes on the sample complexity follow.
>     - Existing sample-optimal schemes are impractical as they require entangled measurements.
>     - Perhaps surprisingly, the optimal sample complexity with incoherent measurements (the setup in our paper) has just been solved very recently, after the submission deadline, in "Tight bounds for state tomography with incoherent measurements" by Chen et al. According to the paper, the optimal sample complexity is $O(D^3 / \varepsilon^2)$ and is achieved by the scheme in ``Low rank matrix recovery from rank one measurements'' by Kueng et al. Nevertheless, as discussed on p. 97 by Kueng et al. in their paper, their scheme is still not practical.
>     - As discussed in Section 1.1 in our paper, [27] is close to being sample-optimal, but empirical results show that the ML approach gives smaller estimation error than their approach. Indeed, "smaller estimation error", as we learned from physicists, is the reason why the ML approach is adopted in practice.

---

### Official Review · Reviewer_o5XD · 2022-07-11

**Rating:** 4
**Confidence:** 5
**Soundness:** 2 fair
**Presentation:** 3 good
**Contribution:** 2 fair

**Summary:**

The submission proposes Stochastic-Q-Soft-Bayes to accomplish the quantum state tomography task. To achieve an $\epsilon$-regret, the required runtime complexity scales with $O(D^4logD/\epsilon^2)$. Numerical simulations on a 4-qubit state demonstrate the effectiveness of the proposed method.

**Questions:**

See the comments above.

**Limitations:**

See the comments above.

**Strengths And Weaknesses:**

Strengths:
The submission addresses an important topic in quantum computing, i.e., devising efficient algorithms to complete quantum state tomography. The paper is well organized and the central idea is well presented.

Weaknesses:
1. It has been proved that $n = \Omega(d^2/\epsilon^2)$ copies are necessary to achieve the estimation error $\epsilon$ [IEEE Trans. Inf. Theory, 63(9):5628–5641, 2017.]. Nevertheless, the submission shows that the proposed algorithm can achieve an $\epsilon$-error using $O(D^4logD/\epsilon^2)$ copies. This contradiction originates from the definition of the `error'. Conventionally, the estimation error between the estimated state $\hat{\rho}$ and the target $\rho$ is defined as $\|\rho-\hat{\rho}\|_{tr}$, whereas the submission adopts the regret to describe the `error'. Consequently, the achieved results are incomparable with prior literature. The authors should clarify this issue.

2. To address the applicability of Stochastic-Q-Soft-Bayes. Simulation results on large-qubit states are desirable. Current results fail to demonstrate the potential advantages of Stochastic-Q-Soft-Bayes.

---

> ### Author Response · Authors · 2022-07-29
> **Inappropriate comparisons**
>
> Dear Reviewer Reviewer o5XD,
>
> Thank your for the comments. If we understand correctly, the two weakness are for comparison to existing results in sample/copy complexity and comparison to theoretically non-rigorous algorithms in empirical convergence speeds, respectively. We address them in the following.
>
> - *``Conventionally, the estimation error between the estimated state $\hat{\rho}$ and the target $\rho$ is defined as $\|\rho-\hat{\rho}\|_{tr}$, whereas the submission adopts the regret to describe the error. Consequently, the achieved results are incomparable with prior literature.''*
>
>     We would like to emphasize that **our aim is not to give a sample efficient scheme, but simply an efficient algorithm to compute the ML estimate.** Yes, as a direct consequence of online-to-batch, we can obtain an algorithm that minimizes the expected negative log-likelihood, of which the minimizer should be exactly the unknown state. Nevertheless, this argument requires random measurements and limits the applicability of Stochastic Q-Soft-Bayes. As Stochastic Q-Soft-Bayes simply aims to minimize the negative log-likelihood, it is simply a numerical optimization algorithm. We had exhaustively compared the optimization speed of Q-Soft-Bayes with the speeds of existing algorithms in Section 3.3 and Section 4.
>
>     One may still want to discuss the distance of an iterate to the minimizer, i.e., the ML estimate, in addition to the error in function value. As the negative log-likelihood is self-concordant, it is locally strongly convex near the minimizer, so the Frobenius norm between an iterate and the minimizer is asymptotically of the order of the square root of the error in function value. We will add this short discussion in the camera-ready version.
>
> - *``To address the applicability of Stochastic-Q-Soft-Bayes. Simulation results on large-qubit states are desirable. Current results fail to demonstrate the potential advantages of Stochastic-Q-Soft-Bayes.''*
>
>     As discussed in Section 3.3 and Section 4, **the two algorithms we chose for comparison in the numerical results either may not converge or lacks a clear complexity bound.** We show the numerical results simply to assess the gap between our *theoretically sound* algorithm and practical *non-rigorous* algorithms. Such a large gap is common in theory works and most theory works just omit such assessments. The value of theory is to point out key ideas and clarify concepts, instead of to give results whose direct applications beat SOTA, so we do not consider the gap a flaw.
>
>     In practice, we may provide the numerical results for the 6-qubit case. We are running the codes on a server. Nevertheless, notice that then the parameter dimension becomes 16 times the dimension for the 4-qubit case and the sample size and per-iteration time grow faster than linearly with the dimension. It seems that the numerical results will not be ready before the author response deadline. We will update the numerical results once we get the new results, if the new results provide more insights.

---

### Official Review · Reviewer_3n1B · 2022-07-11

**Rating:** 5
**Confidence:** 4
**Soundness:** 3 good
**Presentation:** 3 good
**Contribution:** 3 good

**Summary:**

This paper studies state tomography from the perspective of online learning. State tomography is an important problem in quantum computing. Its main objective is to arrive at an approximated description of an unknown quantum state by measuring several copies of that state. The main approach of this paper is based on soft Bayes ML estimation in which the objective is to minimize a log-loss defined between the true density operator and the estimated one. The paper then relates this problem to minimizing regret in online learning (portfolio selection) where soft Bayes is a promising approach. Then, the paper proposes a quantum analogous to soft Bayes. Using classically developed tools, the authors provide guaranteed results for their algorithms. Lastly, they provide numerical demonstrations.

**Questions:**

Why do we need to use online learning for state tomography?

**Limitations:**

The main limitations of the proposed approach are (a) scalability as the number of the iterations grows exponentially with the number of qubits, and (b) numerical experiments do not seem to show the advantage of the proposed algorithm.

**Strengths And Weaknesses:**

The main strength of the paper in my opinion is the quantum soft Bayes which leads to a quantum learning algorithm with provable guarantees. This is a nice idea with possible future applications. The analysis (especially Theorem 3.3) is interesting.

There are two major weaknesses of the paper:

1)  The paper is not clear in the problem formulation.  It is unclear to some extent as to why we need to use the online approach for state tomography where we can have all the samples at once.

2)  The proposed approach does not seem to be scalable as the number of iterations grows exponentially with the number of qubits.

---

> ### Author Response · Authors · 2022-07-29
> **Clarifications**
>
> Dear Reviewer 3n1B,
>
> We thank your for your comments and appreciating our work. We clarify the two weaknesses below.
>
> - *"It is unclear to some extent as to why we need to use the online approach for state tomography where we can have all the samples at once."*
>
>     **We have addressed this in Ln. 38--45.** Perhaps we missed some point familiar to machine learning researchers but not to quantum information experts. Please kindly let us know how we can improve the presentation of that paragraph.
>
>     The reason is the same as why we use stochastic gradient descent (SGD) although we often have the whole dataset in deep learning. The aim is to reduce the dependence of the time complexity on the sample size. Notice that the per-iteration time complexities of SGD and the proposed Stochastic Q-Soft-Bayes are both independent of the sample size. Since the sample/copy complexity for QST is necessarily exponential in the number of qubits, such independence of the sample size is a desired property towards ML estimation for larger quantum systems.
>
> - *"The proposed approach does not seem to be scalable as the number of iterations grows exponentially with the number of qubits."*
>
>     Yes, the number of iteration grows (almost) linearly with the dimension and hence exponentially with the number of qubits. **Nevertheless, as discussed in Section 3.3, the time complexity of the proposed algorithm is already relatively fast compared to existing algorithms that have clear complexity guarantees.**
>
>     Let us discuss whether a weaker dependence of the iteration complexity on the number of qubits is possible. As stated in Ln. 98--100, the optimal regret for OPS is known to be $O ( D \log T )$ and is achieved by UPS. Even if we can ``quantumize'' UPS and the quantumized version inherits the same regret bound, the resulting iteration complexity of "Stochastic Q-UPS," be it possible, will be $\tilde{O} ( \varepsilon^{-1} D )$, still linear in the dimension. Therefore, it is reasonable to expect that we need to sacrifice the dependence on $\varepsilon^{-1}$ in order to get weaker dependence on $D$; then, it seems difficult to claim a fair tradeoff.

---

> > ### Comment · Reviewer_3n1B · 2022-08-08
> > **Thank you for the response**
> >
> >  Thank you for the response. I think the results are interesting, but overall their significance seems borderline. Therefore, I stick to my current rating.

---

> > > ### Author Response · Authors · 2022-08-09
> > > **We would appreciate it if you can explain what is still unsatisfactory to you**
> > >
> > > Dear Reviewer 3n1B,
> > >
> > > We have answered the questions you raised about the problem formulation and you did not point out any insufficiency. We wonder why you still think the significance is borderline. We would appreciate it if you can explain what is still unsatisfactory to you, so we know how to improve on this work.
> > >
> > > Another reason supporting the online formulation can be found in the paper "Adaptive online learning of quantum states", which appears on arXiv after the submission deadline. That paper motivates the study of adaptive regret in online learning quantum states by the following argument.
> > > > Nevertheless, existing literature does not cover an important factor in current quantum computers: fluctuation of quantum states. Currently, many of the state-of-the-art quantum computers, including Sycamore by Google [4], Prague by IBM [32], Zuchongzhi by USTC [39], etc., are superconducting quantum computers. ...In general, fluctuation not only happens in superconducting qubits, but also quantum optical systems including the Rabi model, the Jaynes-Cummings model, etc.; see the textbook [15]. In light of fluctuations, recent study on state tomography extends to temporal quantum tomography [34] with changing quantum states, and from our perspective of machine learning, a fundamental question is to be capable of learning changing quantum states, and this naturally fits into adaptive online learning.
> > >
> > > Whereas that paper focuses on shadow tomography and their Lipschitz loss assumption is not satisfied in our scenario, for state tomography the argument above also holds, motivating us to consider the adaptive regret in online state tomography. To develop an online algorithm with an adaptive regret bound, the standard approach is to introduce a black-box algorithm, e.g., SAOL by Daniely et al. [1], that takes any base algorithm with standard regret guarantee. Q-Soft-Bayes (without "Stochastic") can serve as a relatively computationally efficient base algorithm; the only other existing online algorithm by Zimmert et al. is computationally not scalable w.r.t. the number of qubits.
> > >
> > > If you think the above helps establishing the significance of the online formulation, we will add it in the revision. Thanks!
> > >
> > > ====
> > >
> > > [1] A. Daniely et al. Strongly adaptive online learning. ICML. 2015.
> > >
> > > [2] K.-S. Jun et al. Improved strongly adaptive online learning using coin betting. AISTATS. 2017.

---

### Official Review · Reviewer_m4jW · 2022-07-13

**Rating:** 4
**Confidence:** 2
**Soundness:** 4 excellent
**Presentation:** 4 excellent
**Contribution:** 2 fair

**Summary:**

The paper proposes a quantum version of Soft-Bayes for the maximum likelihood QST task, since Soft-Bayes enjoys fast per-iteration time complexity. The proposed algorithm is a stochastic first order algorithm, and compared to [64], it has better per-iteration complexity on the order of O(D^3), but worse iteration complexity. The paper provides numerical results comparing the proposed algorithm, RrhoR, and Monotonous FW.

**Questions:**

Is it possible to do ML QST when the number of q-bits is large? Why does this problem formulation make sense with exponential dependence on the number of q-bits?

**Limitations:**

Yes.

**Strengths And Weaknesses:**

Strengths
- The paper is clearly written, and the theoretical results are well-supported. The numerical results, though not ideal for the proposed algorithm, are interesting and appreciated.

Weaknesses
- The iteration complexity and per-iteration complexity both involve D, which is exponential in the number of q-bits. Though for the ML QST task this dependence may be inevitable, it would be helpful for the authors to explain why this problem formulation makes sense when the number of q-bits is large.
- The idea of using an online portfolio selection algorithm for a QST task is also explored in [64], and the extension of Soft-Bayes to the quantum setting doesn’t seem to involve new techniques.
- The numerical results are unfortunately not very convincing, as the method obtains higher error than some of the baseline algorithms.

---

> ### Author Response · Authors · 2022-07-29
> **Clarifications**
>
> Dear Reviewer m4jW,
>
> We thank you for your comments. We clarify the weaknesses below.
>
> - *"it would be helpful for the authors to explain why this problem formulation makes sense when the number of q-bits is large."*
>
>     **Scalability with respect to the number of qubits is exactly the motivation of this work.** As we tried to convey in Ln. 35--40, the sample/copy complexity in QST is inevitably exponential in the number of qubits; therefore, batch optimization methods, e.g., gradient descent, must suffer high time costs when the number of qubits is high, as computing the gradient takes time at least linear in the sample size. We hence want an optimization algorithm whose per-iteration time cost is independent of the sample size, just like stochastic gradient descent popular in deep learning. Stochastic Q-Soft-Bayes is one such algorithm.
>
> - *"The idea of using an online portfolio selection algorithm for a QST task is also explored in [64]"*
>
>     **Indeed, an earlier version of this work had been presented in a major quantum information conference that does not publish proceedings almost a year before the appearance of [64] on arXiv.** See also the second footnote in our paper. To follow the double-blind policy, we cannot disclose too much information and do not encourage the reviewers to search for the earlier version of our work on the internet.
>
>     Also, notice that [64] uses Newton's method to compute the iterates, which is computationally prohibitive regarding the high-dimensional nature of the quantum setup (see Ln. 104--110). In comparison, we actually implemented Stochastic Q-Soft-Bayes on a PC and provide a comparison with empirically fast methods.
>
> - *"the extension of Soft-Bayes to the quantum setting doesn’t seem to involve new techniques"*
>
>     As explained in Ln. 63--70, **the technical novelty lies in identifying an appropriate form of the iteration rule; the seemingly easy proof then follows as a merit.** In many quantum information studies, the challenge lies in the fact that there are an infinite number of possible quantum extensions that coincide with the same classical notion, but only one of them inherits the desired property of the classical counterpart, or different quantum extensions inherit different properties. We used the discovery of matrix exponentiated gradient as an example in the paper to illustrate the challenge; other well known examples include quantum extensions of relative entropy, Renyi entropy, f-divergences, etc.
>
>     Identifying an appropriate quantum extension of the iteration rule took us the most amount of time. We had considered several possible quantum extensions, such as $W_{t + 1} = \sqrt{G_T} W_t \sqrt{G_T}$, $W_{t + 1} = \sqrt{W_T} G_t \sqrt{W_T}$, and the one mentioned in Section 5.1. Some of them indeed worked empirically, but we are not able to prove the convergence.
>
> - *"The numerical results are unfortunately not very convincing, as the method obtains higher error than some of the baseline algorithms."*
>
>     Notice that in the numerical result section, **we are comparing the empirical performance of our theoretically rigorous algorithm to those of $R \rho R$, an algorithm that may not converge, and a version of Frank-Wolfe that lacks a clear complexity guarantee.** And we chose them for comparison because they are *fast* in practice; other algorithms involve complicated hessian computations or are obviously slower (e.g., diluted $R \rho R$). Not being able to beat SOTA in practice is common in theory works. **As stated above, the only other online algorithm we know ([64]) is computationally prohibitive; being able to test the algorithm on synthetic data and provide an empirical comparison is indeed a breakthrough.**

---

### Author Response · Authors · 2022-07-30
**Codes updated**

Dear editors and reviewers,

We have updated the codes in the supplementary material. We just noticed that we carelessly put inconsistent versions of the codes in the previous supplementary material. The algorithm implementations are exactly the same; the inconsistency lies in the generation of synthetic data. It is easily checked that the outputs of the updated codes coincide with those in Section 4; notice there is some small perturbation due to the randomness in the synthetic data and algorithm.

The paper and appendix are not changed.

---

### Author Response · Authors · 2022-08-08
**Revision**

Dear Chairs, SACs, ACs, and Reviewers,

We have revised the paper with regard to the review comments.
- Reviewer o5XD asks for numerical results with more qubits and Reviewer m4jW wants to see better numerical results. We managed to optimize our codes such that we can get the results with 6 qubits before the revision deadline. Interestingly, the proposed algorithm becomes faster than $R \rho R$ in terms of the number of epochs. We thank Reviewer o5XD for the suggestion. The superiority of the proposed algorithm should be more obvious with more qubits, but unfortunately our server won't finish the computation before the revision deadline.
- Reviewer oBoV is concerned about the relevance of considering the ML approach to state tomography. In Section 1.1, we explain that sample-optimal schemes are impractical and the only nearly optimal scheme we know empirically has larger estimation error than ML.
- Reviewer 3n1B and Reviewer m4jW wonder why we need an online algorithm when we already have all samples. We tried putting more explanations in the 2nd and 3rd paragraphs of Section 1.

There are many other relatively minor modifications. We color the modified parts in blue so it is easier to find those parts.

We had responded to the review comments more than one week ago, but did not receive any response. We would like to emphasize the following as general statements to the commetns.
- This is a theory work: We value "being provably fast" over "being empirically fast." Otherwise, for example, we would have tried manually tuning the learning rates in the numerical experiment.
- We provided numerical results because *this is the only online algorithm we know that allows for reasonable computational time for the 4- and 6-qubit cases.* This should be considered a significant step toward more scalable ML QST.
- This work focuses on optimization error instead of statistical error. The estimator we consider is just ML; the sample complexity is already determined by the performance of ML. What we consider is simply how to compute the ML estimate in a provably fast way.
- Some reviewers are not satisfied with the exponential dependence on the number of qubits. As discussed in Section 3.3, the overall time complexity of the proposed algorithm is already competitive, if not better, compared to existing ones. As discussed in Section 5.1, if we can find an algorithm whose time complexity is sublinear in $D$, this would immediately imply a big breakthrough for online portfolio selection, a classic open problem since 1991. There are two papers proposing new online portfolio selection algorithms in this year's COLT!
- Sublinear dependence on the number of qubits is a famous feature of shadow tomography. Nevertheless, shadow tomography is not state tomography and state tomography is still an active problem in quantum information.

---

### Meta-Review · Area_Chair_Uvq7 · 2022-08-23

**Recommendation:** Reject
**Confidence:** Certain

**Metareview:**

Overall: The paper propose a stochastic first-order algorithm that computes an -approximate ML estimate for the QST problem.

Reviews: The paper received four reviews. Borderline reject (less confident), Borderline accept (confident), Borderline reject (absolutely confident), Reject (confident). Overall, from the reviews there is not a reviewer that champions the paper for acceptance.

Main issues raised are:
- Clarity of presentation, notation
- Scalability/applicability
- Less relevant to the ML community.

After rebuttal: While the authors have been active responding to the reviewers' comments, the rebuttal discussion was relatively silent. While the AC has reached out to find additional reviewers, this effort was unsuccessful. One of the reviewer was responsive, but the outcome was that the paper still lacks significance and applicability within the ML community. This suggests that it might be preferable the paper be submitted to a near future conference venue (and maybe a more theoretical one), following these suggestions + corrections.

Confidence of reviews: The reviewers are fairly confident in their reviews. The thorough reviews among the four definitely get more weight than the rest of the reviews.

Overall, the paper feels to be in good state but none of the reviewers feels extremely confident championing the paper for acceptance at this venue. We highly suggest the authors to consider near future ML conferences or more quantum-related conferences for resubmission

**Award:**

No

---

### Decision · Program_Chairs · 2022-09-14

Reject